# Development of a collaborative chronic care model for management of cardiometabolic disease in low- and middle-income countries

Pamela Miloya Godia[1*‡], Michelle Hadjiconstantinou[2,3‡], Rosa Weyula[1], Usagi Ememwa[1], Samuel Seidu[2,3,4,5], Peter Njoroge[1], Joyce Muhenge Olenja[1], George Nyadimo Agot[1], Jamin Avugwi[1], Mary Coleman[6], Alfred Yawson[6,7], Filipe Dulce[8], Joselia Chemane[8,9], Celia Novela[8,9], Ana Mocumbi[8,9], Deborah Ikhile[2], Shabana Cassambai[2,4], Albertino Damasceno[8,9], Roberta Lamptey[6,7,10], Kamlesh Khunti[2,3,4,5]

**1** Department of Public and Global Health, University of Nairobi, Nairobi, Kenya, **2** Diabetes Research Centre, College of Life Sciences, University of Leicester, Leicester, United Kingdom, **3** NIHR Biomedical Research Centre, University of Leicester, Leicester, United Kingdom, **4** NIHR Applied Research Collaboration East Midlands (ARC-EM), United Kingdom, **5** Leicester Diabetes Centre, University Hospitals of Leicester, NHS Trust, Leicester, United Kingdom, **6** University of Ghana, Accra, Ghana, **7** Department of Polyclinic, Family Medicine, Korle Bu Teaching Hospital, Accra, Ghana, **8** Universidade Eduardo Mondlane, Maputo, Mozambique, **9** Instituto Nacionall de Saúde, Maputo,Mozambique, **10** Ghana College of Pharmacists, Accra, Ghana

‡ These auhors are joint primary authors on this work.
* pamela.godia@uonbi.ac.ke

## Abstract

### Introduction

Cardiometabolic diseases (CMD) which include cardiovascular disease (CVD), diabetes, hypertension, and other metabolic syndromes represent a significant global health burden. Three quarters of global CVD deaths occur in low-and-middle-income countries (LMICs) and CMD account for approximately 35 percent of deaths in the Sub-Saharan Africa (SSA) region. The COVID-19 Pandemic significantly accelerated the transformation of the landscape in the management of patients with multiple long-term conditions, prompting innovation in healthcare delivery and highlighting the importance of more integrated and adaptable healthcare approaches. Addressing CMD requires a multifaceted approach involving both individual-level interventions, health system approaches, community-based approaches, and broader population-wide strategies for prevention.

### Aim

This study aimed to develop and pilot a person-centred model of health care for CMD management, integrating key principles from the Chronic Care Model (CCM) and Collaborative Care Model (CoCM) to assess feasibility and potential scalability in LMICs.

**Data availability statement:** The paper data is qualitative. The raw data underlying this study are not included with the submission due to ethical and institutional restrictions. Data are available upon request for researchers who meet the criteria for access to confidential data. Kenya data are available upon reasonable request from the Secretary of KNH-UON Ethics and Research Committee via email (Knh-uon_erc@uonbi.ac.ke) or telephone (+254799-495829 / +254799-495830). Ghana data are available upon request from the Director of Medical Services, Korle Bu Teaching Hospital via email (info@kbth.gov.gh) or telephone (+233 302667759). Mozambique data are available upon request from the National Committee of Bioethics for Health, Faculty of Medicine, Maputo Central Hospital via email (cnbs.mocambique@gmail.com) or telephone (+258 824066350).

**Funding:** The research was funded by the National Institute for Health and Care Research (NIHR) under project reference NIHR132995, utilizing UK international development funding from the UK Government to support global health research. The NIHR's Global Health research initiatives aim to fund high-quality applied health research and training in areas of unmet need, primarily benefiting people in low and middle-income countries (LMICs). The funders had no role in the study design, data collection and analysis, decision to publish, or preparation of the manuscript. The views expressed in this publication are those of the authors and do not necessarily reflect those of the NIHR or the UK Government. The NIHR Global Health Research website https://www.nihr.ac.uk/research-funding/global-health.

**Competing interests:** KK has acted as a consultant, speaker or received grants for investigator-initiated studies for Abbott, Astra Zeneca, Boehringer Ingelheim, Bayer, Hikma, Nordisk, Sanofi-Aventis, Servier, Lilly and Merck Sharp & Dohme, Oramed Pharmaceuticals, Pfizer, Roche, Daiichi-Sankyo, Applied Therapeutics, Embecta, Amgen, Bristol Myers Squibb and Nestle Health Science. KK is supported by the National Institute for Health Research (NIHR) Applied Research Collaboration East Midlands (ARC EM), NIHR Global Research Centre for Multiple Long-Term Conditions, NIHR Cross NIHR Collaboration

## Methods

The development of the CREATE intervention took a mixed method approach utilizing both qualitative and quantitative methodologies, including a systematic review, qualitative synthesis, and needs assessment including the delivery of workshops with local stakeholders and people living with CMD in Ghana, Kenya and Mozambique.

## Results

A CoCCM with the following components was developed as the CREATE intervention: 1) Self-Management support, 2) Decision support (which included health care provider training), 3) Community linkages, 4) Organisation of health care, 5) Clinical information system, and 6) Delivery system design (streamlining the referral pathway). The CREATE intervention was informed by a systematic review, needs assessment, and six stakeholder workshops across three LMICs, identifying barriers such as limited primary care infrastructure, lack of referral systems, and gaps in self-management education.

## Conclusion

This is the first CoCCM model for Multiple Long-term Conditions (MLTC) to be developed for SSA. The intervention is currently being tested as part of a feasibility study in Kenya, Ghana and Mozambique. The CREATE intervention has the potential for adaptability to local context, however there is need for more rigorous research to evaluate the model effectiveness in relation to improving patient outcomes.

## Introduction

Cardiometabolic diseases (CMD) which include cardiovascular disease (CVD), diabetes, hypertension, and other metabolic syndromes represent a significant global health burden. Globally, CVDs are one of the leading causes of death, responsible for approximately 179 million deaths per year, accounting for 31 percent of all global deaths. Three quarters of global CVD deaths occur in low-and-middle-income countries (LMICs) and CMDs account for approximately 35 percent of deaths in the Sub-Saharan Africa (SSA) region [1]. There has been an increase in the burden of non-communicable disease (NCDs) in Sub-Saharan Africa (SSA) which is largely influenced by a complex interplay of environmental, social, political, and commercial factors with 80 percent of the 463 million people living with diabetes worldwide living in LMICs [2]. In SSA specifically, between 4–5 percent of adults have Type-2 diabetes (T2D), with the majority remaining undiagnosed, and 19–31 percent of adults are overweight. Additionally, over 1·1 billion people worldwide have hypertension, with two-thirds residing in LMICs [2]; hypertension, if left untreated, can lead to a spectrum of CVDs. Africa remains to have the lowest levels of awareness, treatment and control of blood pressure worldwide [3] thus decreasing the life expectancy.

for Multiple Long-Term Conditions,NIHR Leicester Biomedical Research Centre (BRC) and the British Heart Foundation (BHF) Centre of Excellence. SS is in receipt of speaker honoraria from AstraZeneca, Boehringer Ingelheim, Janssen, Lilly, MSD, Abbott, Novo Nordisk, SB Communications, OmniaMed Communications, Roche, Napp Pharmaceuticals, NB Medical and Amgen; advisory board honoraria from AstraZeneca, Lilly, Boehringer Ingelheim, Janssen, Abbott, MSD, Novo Nordisk, Takeda and Sanofi; educational grants from Boehringer Ingelheim, Lilly, Novo Nordisk, and Takeda; and conference registration and subsistence from Boehringer Ingelheim, Janssen, Lilly, Novo, Nordisk, Abbott and Takeda. RL has received speaker honoraria, advisory board or educational grants or research funding: Novo Nordisk, Boehringer Ingelheim, AstraZeneca, Sanofi. SC is supported by the National Institute for Health Research (NIHR) Global Health Research Group for Cardiometabolic Disease Research in Africa Partnership (CREATE). SC, DI, are supported by the NIHR Applied Research Collaboration East Midlands (ARC EM), NIHR Global Research Centre for Multiple Long-Term Conditions. PG, MH, RW, UE, PJ, JO, GA, JA, MC, AY, DF, AM, JC, CN, AD have declared no competing interests exists." "This does not alter our adherence to PLOS ONE policies on sharing data and materials."

**Abbreviations:** CMD, Cardiometabolic Diseases; CVD, Cardiovascular Diseases; LMICs, Low-and-Middle-Income Countries; SSA, Sub-Saharan Africa; CCM, Chronic Care Model; CoCM, Collaborative Care Model; CoCCM, Collaborative Chronic Care Model; CREATE, Cardiometabolic disease research in Africa partnership; SME, Self-Management Education; HCPs, Health Care Providers; MLTC, Multiple Long-term Conditions; T2D, Type-2 diabetes; NCDs, Non-communicable disease; HIV, Human Immunodeficiency Virus; TB, Tuberculosis; HICs, High-income countries; DALYS, Disability-Adjusted Life Years; HbA1c, Glycated haemoglobin; DSMES, Diabetes self-management education and support; BCTs, Behaviour change techniques; NICE, National Institute of Health and Care Excellence; PBA, Person-based approach; SEG, Socio-economic groups; CEI, Community Engagement and Involvement; MRC, Medical Research Framework.

The complex reasons for the increase in NCDs and CMD include: urbanisation with negative lifestyle such as less physical activity, higher calorie diets, high prevalence of smoking, increase in ageing population, underfunded healthcare systems with a major focus on communicable diseases (e.g., HIV, TB, and Malaria), inadequate access to healthcare including screening and self-management education programmes, and complex socio-cultural belief systems [4,5].

CMD have a differential trend in prevalence and mortality between high-income countries (HICs) and LMICs. While CVD mortality has decreased in HICs due to improved access to healthcare, it has risen in LMICs due to a combination of under-resourced healthcare systems, late diagnosis, and poor risk factor control. Diabetes prevalence has also increased more rapidly in LMICs, with the highest burden in sub-Saharan Africa (SSA). In SSA, age-standardized DALYs are 44 percent higher among lower-income groups, highlighting significant healthcare disparities [1,6]. However, data published in the World Health Organization Mortality Database indicate that the rate of decline in mortality due to cardiovascular disease in some high-income countries has slowed down or plateaued particularly for persons aged 35–74yrs; for example, North America (USA and Canada), mortality rates due to CVD have increased in recent years [7]. In Switzerland, while CVD mortality continues to decrease in most of the population, specific age groups, such as those aged 60–74, have seen a plateau in stroke and coronary heart disease mortality rates. The plateau could be attributed to several factors such as lack of regular physical activity, unhealthy diets, high alcohol consumption, persistent smoking and social inequalities among different segments of the population [8].

Although CMD in LMICs have often been reported to occur among wealthier persons, a faster increase trend is being observed in poorer populations. Socioeconomic status significantly influences access to healthcare, lifestyle choices, and exposure to risk factors such as poor diet and lack of physical activity. The "capacity-load" model has been useful in understanding CMD in LMICs and outlines the following as important: nutrition, nutrition environment, air pollution, urbanisation, and commercial determinants of health [9]. These inequities further disadvantage the urban poor, who are most likely to reside in informal settlements lacking essential health infrastructure. Some infectious diseases increase the risk of developing NCDs. In sub-Saharan Africa, 11·7 percent of crude NCD burden is attributable to infections, a proportion that is higher than the common risk factors for NCDs [10]. There is therefore a synergistic relationship between infectious and NCDs where the presence of one condition can worsen the severity and progression of others ultimately affecting the overall health outcome. Social and environmental determinants of health shape these synergistic epidemics.

## COVID-19 and metabolic diseases

The COVID-19 pandemic accelerated a shift toward integrated and remote care models, including the adoption of telehealth and self-management strategies, which significantly influenced the management of patients with multiple long-term conditions [11]. This stemmed from the fact that during the COVID-19 pandemic, prevention

measures put in place by governments such as social distancing and isolation led to the closure of some hospitals and non-emergency cases were asked to stay at home or given longer appointments as they were regarded not urgent [12,13]. This had an overall effect of increasing health inequalities including reduced access to healthcare and provider consultation, reduced adherence to medication and access to other essential primary health care services [14].

## Management approaches for CMD in LMICs

Addressing CMD requires a multifaceted approach involving both individual-level interventions, health system and policy level approaches, community-based approaches, and broader population-wide strategies for prevention. These approaches have the potential to strengthen the capacity of health systems to manage the growing burden of CMD in LMICs. While traditional approaches to CMD in LMICs have focused on risk factor identification, reduction and treatment, there is a growing emphasis on upstream prevention strategies targeting entire populations [15]. These population-wide interventions aim to shift the distribution of risk factors across large groups. For instance, increasing access to high blood pressure treatment by 70%, reducing sodium intake by 30%, and eliminating artificial trans fatty acids could potentially prevent 94 million deaths globally within 25 years [16]. Peru's recent implementation of a salt-substitution strategy exemplifies this approach, leading to community-wide reductions in blood pressure and hypertension [17], while in the Chinese study salt substitution resulted in lower rates of stroke and major cardiovascular events [18]. Population level interventions have also been used in smoking cessation programs [19], obesity and metabolic diseases [20]. LMICs face unique challenges in providing long-term, patient-centred care required for CMD as the already stretched healthcare systems are often geared towards acute illnesses, maternal and child health, and infectious diseases [9]. mHealth technologies and integrated care are some of the emerging innovative solutions that can bolster the health system in CMD management [21]. Community-based health care for CMD in LMICs has heavily relied on task-shifting, leveraging non-clinical workers and providing them with pre-requisite knowledge and equipping them; as a cost-effective alternative to specialist care [15]. These initiatives, coupled with equipping Community health workers with household-level screening and referral tools, strengthen community-based CMD management [22]. Importantly, training non-physician health workers as CMD health educators empowers patients through self-management education [23]. This approach has a wider reach in the community and has proven to enhance screening and health education for non-communicable diseases [24]. Individual Care Approaches such as peer-peer support has shown to be promising in helping individuals cope with negative emotions and navigate barriers to disease management [25].

## Key principles of chronic care model and collaborative care model

Chronic Care Model (CCM) is a well-integrated model for management of chronic diseases in HICs and comprises the key elements of a health care system that encourage high-quality chronic disease care (Table 1). The key components of CCM include the community, the health system, self-management support, delivery system design, decision support, and clinical information systems [26]. It has a care manager responsible for care coordination, shares care management with the patient, and provides the platform for ongoing collaboration between different care providers. Wagner's CCM outlines intervention areas for the effective management of patients with chronic illness which include the use of evidence-based, planned care; reorganisation of practice systems and provider roles; improved patient self-management support; increased access to expertise; and greater availability of clinical information [26,27]. Additional CCM elements identified as important in LMIC include a focus on structured self-management education; quality of communication between healthcare providers and patients; continuity of care; raising awareness of the availability of essential medicines, diagnostics, and trained personnel at primary levels of healthcare; and the use of community groups [28]. However, the challenge remains on how to effectively integrate all these components into primary health care settings in LMICs with very little evidence base. Evidence from HICs shows that CCM is effective in improving biomedical outcomes, including glycated haemoglobin (HbA1c), blood pressure, creatinine levels, cholesterol levels, and fasting glucose; enhancing patient-reported

**Table 1. Differences between chronic care model and collaborative care models.**

| Aspect | Chronic Care Model (CCM) | Collaborative Care Model (CoCM) |
|---|---|---|
| Focus | Systematic approach to restructure health care systems for chronic disease management. | Integrated care approach focusing on collaboration among health care providers. |
| Key Elements | Health system organization, self-management support, decision support, delivery system design, clinical information systems, and community resources and policies | Primary care providers, care managers, and behavioural health specialists working together. |
| Patient Involvement | High emphasis on patient self-management and education | Emphasis on patient-centred care with active involvement of care managers |
| Implementation | Requires significant changes in health care infrastructure and policies | Focuses on integrating existing resources and enhancing provider collaboration |
| Effectiveness | Shown to improve clinical outcomes like HbA1c and blood pressure control | Effective in improving mental health outcomes and overall patient satisfaction |
| Challenges in LMICs | Limited resources, lack of trained personnel, and inadequate health care infrastructure | Similar challenges, with additional barriers in integrating mental health services |
| Gaps | A robust health care systems and policies to support chronic care in primary care settings | Integration with mental health services and training for primary care providers |

outcomes, such as health-related quality of life; and reducing diabetes complications following the implementation of one or more of its elements. Additionally, literature also highlights the importance of incorporating evidence-based decision-making throughout the treatment and management process in a CCM [28].

The Collaborative Care Model (CoCM) is an evidence-based approach that integrates care services to better manage chronic diseases. The model integrates the use of evidence-centred care from the Chronic Care Model [29,30]. The goal is to reduce costs, de-stigmatize the illness, promotes one-on-one consultation, increase access to services, and improve patient outcomes. Though reported to be effective, evidence for its use in LMICs is limited [31]. There are few examples of CCMs or CoCMs in LMICs. There are bottlenecks around how to implement and scale up a CCM/CoCM approach in LMICs and these include methodological limitations, lack of standardized data collection, limited trans-disciplinary research, and lack of translation of research findings to action. Patients with MLTC experience a range of challenges, including limited services at the primary healthcare level; lack of essential medicines, particularly in public health facilities; and limited access to laboratory services. This study aimed to develop a person-centred model of health care for management of CMD based on key principles of the CCM and CoCM.

## Adaptations to the traditional chronic care model

Recent literature highlights substantial adaptations to the traditional CCM, aimed at strengthening cross-sector collaboration and integration in chronic disease management. These revisions emphasize multidisciplinary teamwork, leveraging technology, systemic reforms, and patient-centered and holistic care, ensuring a more holistic and coordinated approach.

Multidisciplinary teams (MDTs) integration now forms the backbone of the revised CCM, incorporating pharmacists, nurses, general practitioners, and psychologists to deliver comprehensive, patient-centered care [32,33]. The inclusion of pharmacists improves medication adherence and reduces errors, enhancing outcomes while reducing healthcare costs [32]. The integration of digital health tools, eHealth platforms, and interoperable information systems strengthens communication and data sharing across sectors [33,34]. Digital solutions enhance coordination of patient care by enabling proactive monitoring, population-level data analysis, and have also been critical in the integration of mental and physical health management [35]. The eHealth Enhanced CCM (eCCM) highlights the value of virtual communities and feedback loops in boosting patient engagement [36]. Recent literature calls for policy reforms, shared care plans, and interprofessional training to standardize collaboration and clarify roles and responsibilities within MDTs [33]. Revisions of the CCM include integration of physical, emotional, and social dimensions of health, hence fostering individualized care plans that

improve satisfaction and outcomes. The Expanded Chronic Care Patient–Professional Partnership Model (E2C3PM) is a novel integrated model that leverages patient experience and knowledge, mediation, therapeutic education, and partnership to empower patients, rebalance power dynamics, and foster a more comprehensive and collaborative approach to chronic disease management, leading to improved care integration and outcomes [37] Community integration across healthcare and social sectors improves care coordination for complex patient needs, reducing exclusion and supporting continuity of care. Applications of the CCM in contexts such as the Veteran's Health and correctional systems demonstrate its adaptability to diverse populations and institutional settings [38]. Community-based organizations and primary care nurses are increasingly recognized as key actors in supporting self-management and extending care beyond clinical settings [39].

Despite these advancements, barriers such as resistance to change, lack of standardized protocols, and limited adoption of eHealth and patient-centred and partnership approaches persist [34]. Effective implementation requires sustained innovation, organizational cultural shifts, and adaptation to local contexts to fully optimize chronic care delivery.

## Chronic care model and collaborative care models for people with multiple long-term conditions

Multiple long-term conditions (MLTC) or multimorbidity often refers to the coexistence of two or more chronic diseases in one individual. Across many LMICs, there is limited research on NCDs, particularly on addressing multiple long-term conditions or shared risk factors. Most studies tend to focus on single NCDs. This is a significant gap considering there is evidence showing a rising trend in the prevalence of MLTCs in LMICs. Whilst there are not many examples of CCMs or CoCMs in LMICs, an effective example from the USA is a randomised control trial among people with depression co-existing with poorly controlled diabetes, coronary heart disease, or both where positive health outcomes were reported in the intervention group [40]. A similar model was successfully adapted for people in India with depression and diabetes [41]. However, there is still demonstrable evidence gaps around how to implement and scale up a CCM/CoCM approach in LMICs. Significant methodological limitations, lack of standardized data collection and limited trans-disciplinary research also result in lack of translation of research findings to action. Consequently, there is a demand for effective efficient sustainable management options for multiple long-term conditions including CMD. A systematic review and meta-analysis of multimorbidity of NCDs in LMICs has shown a higher risk of multimorbidity associated with age, sex, being well-off, and urban residence with cardiometabolic and cardiorespiratory conditions being the most common multi-morbidity patterns [42]. This approach directly aligns with many priorities outlined in the World Health Organization Global Action Plan for the Prevention and Control of NCDs [43]. There exists knowledge gap in the adaptation, scalability and sustainability of chronic care models that have been successfully implemented in high income countries to fit the LMIC context in relation to constraints related to the health system, infrastructure and social and cultural factors. To date, evidence on the implementation of collaborative care models emanate from high income countries with little adaptation to LMIC settings [44–46]. There is therefore a lack of evidence on how to effectively adapt and scale interventions in diverse LMIC settings, which is crucial for addressing common chronic conditions such as CMDs and mental health [45,47]. Many LMICs face health system constraints such as inadequate infrastructure, shortage of trained health professionals, limited access to essential medications and diagnostic tools [48], which may pose significant challenges in the implementation of collaborative care models. This calls for the need to develop and test novel low-cost approaches which are tailored to resource-limited settings to address these disparities [48]. There is also knowledge gap in the understanding the use of digital technology, health care provider training, and supervisory structures including how to effectively integrate mid-level practitioners and community health workers necessary for effective chronic care management [49]. The CREATE project's CoCCM intervention proposes integrating CCM's system-level strategies with CoCM's patient-centred, team-based care approach to enhance health outcomes and improve care delivery for CMD in LMICs.

## Challenges and innovations in CMD management

Management of CMD in Kenya, Ghana and Mozambique face challenges which include health system factors, social and cultural factors such as lack of access to early diagnostic tools, lack of access to essential medicines and frequent stockouts in public hospitals, use of traditional medicine and disproportionate distribution of health care providers between urban and rural settings. For instance, Kenya has reported an increase in the prevalence of hypertension, but only 4% of the patients under treatment achieve control [50]. With regards to early detection of diabetes, more that 88% of Kenyans have never had their blood glucose tested, leading to late diagnosis [50]. Less than 50% of hospitals in Kenya have the equipment and trained staff required for essential diagnostic procedures like electrocardiography and ultrasonography and only 49% of hospitals have a complete stock of necessary medications, with frequent stock-outs reported [48]. Similarly in Ghana challenges include inadequate infrastructure, workforce shortages, fragmented coordination of care, high costs of care and out-of-pocket expenditure, and governance inefficiencies for chronic disease management, reliance on traditional medicine, cultural beliefs and delayed care seeking [51,52].

Promising innovations in the management of CMDs in Kenya, Ghana and Mozambique include community-based lifestyle interventions which have reported significant reductions in metabolic syndrome markers such as improved diet and exercise adherence [53], and task-sharing models focusing on community health worker-led screening and linkage programs leveraging on digital technology [54]. The *Akoma Pa* digital, faith-based pilot intervention tested in Ghana has reported higher patient follow-up rates, improved blood pressure and glycemic control, and demonstrated favorable cost-effectiveness [55].

## The CREATE project

CREATE stands for Cardiometabolic disease REsearch in Africa parTnErship. This is a multi-country feasibility study that is being implemented in Kenya, Ghana, and Mozambique with the aim of improving risk factor control, improving care and outcomes for patients with cardiometabolic multiple long-term conditions using a CoCCM of care. The CREATE project is being implemented in four work packages whose outcomes are scoping and needs assessment, intervention development, carrying out a feasibility study and impact and sustainability.

The chronic conditions focused on by this program are type 2 diabetes mellitus (T2DM), hypertension, cholesterol, and cardiovascular disease. Our decision to focus on these four conditions was informed by evidence showing an increase in the cardiovascular morbidity and mortality in LMICs, the need to improve patient outcomes, and our clinical knowledge and global research expertise. These are also the commonest cardiometabolic clusters of MLTCs. Between 1990 and 2019, there has been a 131.7% increase in CVD prevalence [56] which compounds cardiometabolic disease burden as it increases all-cause mortality risk by 4–7 fold [57]. Cardiometabolic conditions often co-exist in an individual making an integrated care approach more efficient.

There is an interconnectedness of these four chronic conditions which exacerbates morbidity and mortality, making it crucial to address them collectively. These conditions are prevalent and contribute to the high morbidity and mortality in these countries. Type 2 Diabetes Mellitus (T2DM), hypertension and dyslipidemia are significant risk factors for cardiovascular disease, but prevention and management are suboptimal due to inadequate healthcare systems and lack of resources. For example, hypertension remains the major risk factor for stroke and hence also a major contributor to the onset of CVD [58]. Persons living with T2DM are 2–3 times more likely to experience cardiovascular events [59]. In addition, hypertension is twice as common among people with diabetes, with up to 75% of cardiovascular disease in diabetes attributable to hypertension [60].

## Aim

This paper focuses on the intervention development work package of the study whose aim was to develop a person-centred model of health care for management of cardiometabolic disease.

## Specific objectives

The specific objectives are:

1. To identify and prioritize healthcare needs and available resources to support model development.

2. To iteratively develop a model for people with CMD, implement and evaluate the model in Kenya, Ghana and Mozambique.

3. To investigate the practical implications of delivering the model developed.

## Methodology

The development of the CoCCM under the CREATE project took an iterative process that employed a mixed-methods approach This process involved review of literature, identifying and prioritising the healthcare needs of people living with CMD; assessing available resources to support healthcare delivery and model development; and, based on these findings, iteratively developing a care model for CMD management, ready for implementation in SSA and other LMIC.

To develop the bespoke CoCCM for the CREATE project, we first identified the needs of people living with CMD, as well as health system and community factors influencing their care. This was achieved by conducting a scoping exercise of existing literature on models of care for people living with multiple long-term conditions, focus group discussions as well as holding a series of stakeholder workshops. The implementation feasibility of the CREATE intervention was assessed through structured pilot testing across study sites. Results of the feasibility study will be published in a separate paper.

### Quantitative approach

Identifying and prioritizing health needs and available resources required for model development was done through a series of activities which included a systematic review and scoping analysis. A systematic review was conducted to identify the magnitude of cardiometabolic diseases in Sub-Saharan Africa with a focus on CCM and CoCM. Details of the systematic review have been published elsewhere [61]. In summary, the results from the systematic review showed that the most widely studied conditions were type 2 diabetes, hypertension, stroke, and cardiovascular diseases, including hypercholesterolaemia. These results highlighted key areas to consider during the intervention development phase, such as the definition of hypertension as blood pressure above 140/90 mmHg; the definition of type-2 diabetes as fasting plasma glucose greater than 126 mg/dL (>7·0 mmol/L), random blood glucose greater than 200 mg/dL (>11·0 mmol/L), or HbA1c of 6·5% or higher; and the estimated pooled prevalence of hypertension (27·1%), type 2 diabetes (5·9%), hypercholesterolaemia (10·8%), cardiovascular disease (7·0%), and stroke (1·6%), highlighting the need for targeted models of care for CMD in sub-Saharan Africa [61]. These findings informed the standardization of the definition of the cardiometabolic disease across the three study sites.

### Qualitative approach

A scoping analysis and needs assessment were thereafter conducted through focus group discussions and stakeholder workshops whose aim was to identify the needs, key characteristics and general health status of persons living with CMD and the impact of CMD on patients' lives. The workshops also helped identify national priorities and strategies with respect to CMD; describe the current referral pathways for patients with CMDs; identify barriers and enablers to care for persons with CMD; and explore the local and cultural factors affecting the general health status of communities. A total of 6 stakeholder workshops were held in the three countries Kenya (2), Ghana (2) Mozambique (2). The workshops took place in the study sites and at community halls and venues and were facilitated by the research team in each country.

The qualitative interviews and stakeholder workshops revealed several key unmet needs among people living with CMDs. These included: lack of knowledge of the signs and symptoms of CMDs leading to poor health-seeking behaviour

and late diagnosis, prevalence of stigma, misconceptions and fatalism, inadequate primary healthcare infrastructure; limited availability and accessibility to essential medicines, inadequate equipment and resources at health facilities, especially at the primary health care facilities, lack of access to health insurance; and lack of political commitment to the management of CMD. Patients also cited high costs of laboratory monitoring, unhealthy dietary practices and poor health-seeking behaviour. For example, care-seeking was often initiated by an acute episode such as collapse. Many patients wanted to treat CMDs as acute illnesses and often asked if they could discontinue taking medication if they got symptom relief. Additionally, there was a notable lack of health education and suitable educational materials to support disease management and awareness.

With regards to available resources, there were existing national policies and guidelines in the three countries which provided a framework for intervention design. However, there were notable gaps in the implementation and weak referral practices. Stakeholders also cited negative health care provider attitude, provision of fragmented care and the need to train healthcare providers to improve communication among themselves and with patients. At the community level, the following community structures were identified for intervention delivery: patient support groups, community health workers and volunteers, community leaders, women groups, church groups and keep-fit clubs. These perceptions informed the design of the SME, the community linkages components of the intervention and the training packages for health care workers and community champions.

With regards to intervention delivery stakeholders and persons with lived experiences noted that the intervention should be designed in such a way that it is delivered at certain venues, days, time to address challenges around transport and work-life commitments. It also transpired that literacy was a unique need across the study settings. This prompted the development team to develop participant resources including patient facing materials that were user-friendly, visually-led (less text-heavy) and culturally relevant. This key information informed the model development as well as the design of the Theory of Change.

During the stakeholder workshops, the following strategies were recommended for improving care for persons with MLTCs. These were considered during the intervention development process: multi-level health education targeted at patients, community members and healthcare providers; prioritising and leveraging on existing community resources to deliver trusted and culturally appropriate interventions for CMD management; providing universal health coverage underpinned by principles of accessibility, affordability, patient-centredness and accountability; and taking a collaborative approach to care and management.

The development of the model was also informed by previous work the research team had conducted under the diabetes self-management education and support (DSMES) programme in Malawi, Mozambique [62] and Ghana [63,64]. This work particularly shaped the Self-Management Education (SME) component of the CREATE intervention.

Ethical Approval for the study was obtained from Research and Ethics Committee from the three participating countries. In Kenya, ethical approval was obtained from Kenyatta National Hospital – University of Nairobi (KNH-UON) Ethics and Research and Committee. In Ghana ethics approval was obtained from Institutional Review Board Korle Bu Teaching Hospital (KBTH), and in Mozambique, ethics approval was obtained from the Institutional Committee of Bioethical in Health, Faculty of Medicine, Maputo Central Hospital. All participants provided written informed consent witnessed by the research team.

Recruitment of study participants who took part in the CREATE feasibility study in the three countries started on 14th August 2023 and the end date was 30th August 2024.

### Theory of change for CREATE intervention

The CREATE programme was guided by the Theory of Change framework developed during the early stages of the research project (Fig 1). Theory of Change was fundamentally driven by the impact we wished to achieve and visually depicts why we thought CREATE would be effective, considering not only individual determinants, but also societal,

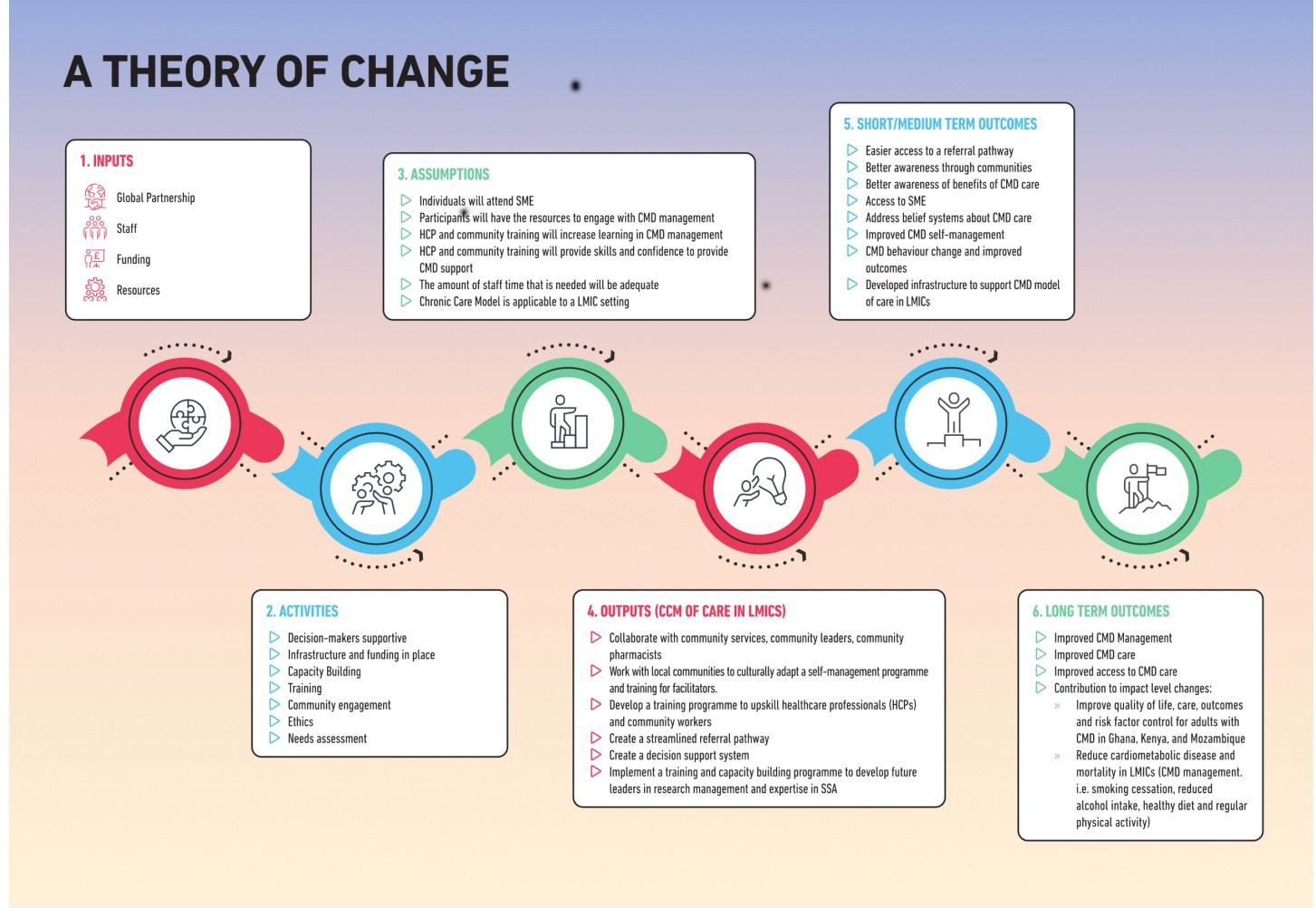

**Fig 1. Theory of change for CREATE prototype.**

financial and policy factors. The prescribed theory of change required that we are participatory in our approach to attain an intervention that is easily implementable, realistic, and tailored to the needs of our participants. Theory of Change framework suggested four key features for inclusion in the CREATE intervention: (i) credible, based on the team's and stakeholders' previous experience and based on existing literature; (ii) achievable, and include appropriate resources to develop the intervention; (iii) supported by local stakeholders, including beneficiaries, frontline staff etc; and (iv) testable, including a detailed description of the intervention and outcome indicators in order to collect the relevant data for the feasibility study.

## Behaviour change techniques

In addition to the needs assessment described above, we also carried out scoping literature search to identify the most common and effective behaviour change techniques (BCTs) needed to improve the targeted behaviours for our study population. The BCTs described the active ingredients of the intervention. For example, a recent systematic review of randomised controlled trials assessing the effectiveness of mHealth interventions to improve taking statin medication,

identified "Goal setting (behaviour)", "Instruction on how to perform a behaviour", "Credible source", "Information about health consequences", "Feedback on behaviour", and "Social support" as the most common BCTs used in effective interventions [65].

A multi-European project conducted in 2021 involving 12 countries in the development of a behaviour change competency framework to support behaviour change in self-management of chronic disease suggested a core set of 21 BCT linked to diet, physical activity and medication use in seven long-term conditions which included type 2 diabetes, COPD, obesity, heart failure, asthma, hypertension, and ischemic heart disease. This is of a particular relevance as the identified conditions from this study match the target conditions in our study [66].

The above BCTs largely align with the National Institute of Health and Care Excellence (NICE) guidelines (2014) for designing individual-level behaviour change interventions aimed at promoting behaviour change in modifiable risk factors associated with CVD and type 2 diabetes. Recommendations in these guidelines suggest that interventions and programmes utilising goals and planning, feedback and self-monitoring (behaviour and outcomes), and social support are likely to be the most effective at promoting behaviour change in lifestyle behaviours associated with CVD and T2DM.

Triangulation of evidence from literature synthesis and needs assessment informed the selection of key intervention components and behaviour change techniques to help elicit behaviour change. Keeping in mind the local cultural context and linguistic differences across the three LMIC countries, the intervention was adapted for each country and community. Although the work focused on the adaptation of the intervention to each study site, the process required extensive planning to develop content specific to CMD management in LMICs. To ensure that the SME sessions were engaging, meaningful and relevant to the target audience, the person-based approach (PBA) model was adapted [67]. A key element of the PBA model which ensures that the intervention is feasible and effective in terms of the behavioural context, is the guiding principles which addressed the following three questions: What is the aim of the intervention in relation to behaviour change? What are the characteristics of the target users (psychosocial characteristics and the behavioural context)? What behavioural needs or issues will the SMEs address? (Table 2).

## CREATE intervention model components

Based on the collaborative chronic care model, the following components were developed to facilitate the implementation of the CREATE intervention (Fig 2). These included:

1. Self-management support. This comprises of a structure self-management education (SME) programme for clients and a training package for educators to build their skills and knowledge to deliver the SME

2. Decision support: This consists of a training package to upskill Health Care Providers (HCPs) to improve knowledge base, decision making and motivational interviewing in managing cardiometabolic diseases

3. Community linkages: this comprises of a training package to educate community leaders including persons living with cardiometabolic diseases as champions about CMDs.

4. Organisation of health care: this includes supporting improvement in service delivery at different levels of the health system through training programmes and coordination of care.

5. Clinical information system: A decision-making tool to initiate the implementation of a data registry

6. Delivery system design: this includes streamlining the referral and care pathways

Each of these components were culturally adapted to the three LMICs that participated in the study by engaging local community members through the establishment of Community Engagement and Involvement (CEI) panels. The CEI panel was composed of patients, health care providers, health educators, community leaders, religious leaders, herbalists, caregivers, and fitness instructors. The role of the CEI was to contribute to a better understanding of cultural and contextual

**Table 2. Steps guided by the person based approach (PBA) model.**

| What is the aim of the intervention in relation to behaviour change? |
| --- |
| Reduce complications of cardiometabolic disease (type 2 diabetes, cardiovascular disease) |
| Improve CVD/T2DM knowledge and risk factors? (blood pressure, cholesterol, smoking, dietary behaviour, physical activity, alcohol consumption, weight, Body Mass Index, and waist circumference) |
| Encourage healthy eating such as reduced intake of salt, sugar and fat |
| Increase physical activity, increase muscle/bone strength and reduce sedentary behaviour/sitting time |
| Improve medication adherence |
| Improve mood and wellbeing (emotions, self-help, relaxation techniques, stigma) |
| Reduce stigma associated with CMD |
| Recognise their own physical/emotional health and seek medical attention when needed |
| What are the **characteristics** of the target users (i.e., psychosocial characteristics and the behavioural context)? |
| Adults with CMD (diabetes, hypertension and cardiovascular disease) who will attend an SME programme to improve lifestyle behaviours, which comes under a wider chronic care model for CMD in Africa. Adults with CMD may have the following behavioural characteristics: |
| Limited education and low health literacy (cannot read or write English, Portuguese or local dialect) |
| Limited understanding and knowledge of CMD |
| Limited skills and confidence for CMD management behaviours |
| Lack of affordability of medication, or healthy food |
| May live in rural areas with minimum resources |
| Must travel far to get to a hospital |
| Rely on traditional medicine |
| What are the **behavioural needs or issues** that the SMEs will address? |
| Lack of understanding of food portion control |
| High salt intake is common |
| High saturated fat diets are common |
| Consumption of traditional high calorie foods is common |
| Consumption of fast-food or commercially prepared food |
| Low socio-economic groups rely on carb-heavy diets because of costs compared to vegetables |
| Recreational physical activity is a problem for patients |
| Sedentary lifestyle by high socio-economic groups |
| Perception that traditional medicine is cheap, readily available and culturally acceptable |
| A lot of trust and credibility given to religious/ spiritual practices |
| Hold beliefs that witchcraft can treat diabetes |
| Hold negative perception of insulin |
| Hold negative perception of body weight and obesity |

problems relating to diagnosis, management and research of CMDs, provide feedback on the intervention design and implementation and contribute to the creation of support systems for future research opportunities.

1. **Self-management support:** In line with the recently updated Medical Research Framework (MRC) on programme adaptation, we adapted an existing SME programme specifically for type 2 diabetes management, previously tested in Malawi and Mozambique [68]. This was supported by the development team at the Leicester Diabetes Centre, UK. The SME programme, consisting of 15 sessions spread over multiple days, was designed for interactive learning, ensuring participants could fully engage with concepts and practical applications. The SME programme is an evidence-based and theory-based SME programme, delivered in a group-setting, and covers CMD interactive topics including

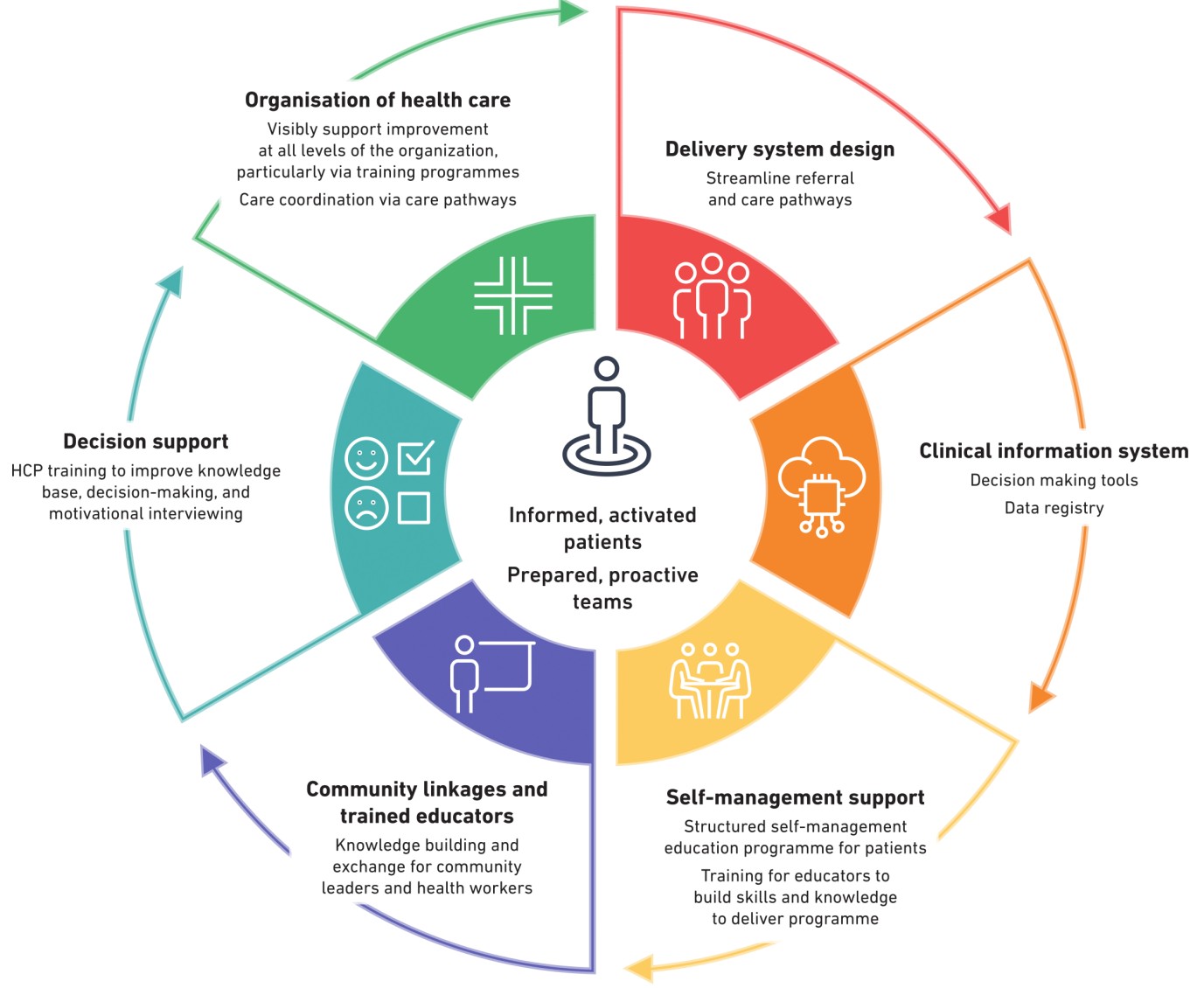

**Fig 2. The CREATE collaborative chronic care model conceptual framework.**

understanding Type 2 diabetes, hypertension, cholesterol and cardiovascular disease. The sessions also cover strategies for self-management such as monitoring my health, diet and food portion control including salt intake, understanding medication, physical activity, mood and well-being, and effects of smoking and alcohol consumption.

Following the adaptation of the SME prototype, further feedback was obtained to gain insight into patients' impression of the SME. This included sharing the philosophy, curriculum and accompanied resources with persons living with CMD and healthcare professionals in local communities. Feedback was focused on elements of intervention engagement, acceptability, usability and delivery as part of the feasibility study. The SME program was rigorously tailored to the local contexts of the countries involved, ensuring cultural and linguistic appropriateness. Cultural adaptations included the use of locally sourced food examples, such as cassava and plantain in Ghana and ugali in Kenya, alongside culturally relevant

visual aids to illustrate balanced diets. For example, in Kenya, pictures of local fruits, staple foods, types of meat and fish as well as a diversity of traditional vegetables were used to demonstrate food portions. To address the barriers to low levels of health literacy, educational materials were translated into Portuguese for Mozambique, Swahili for Kenya, Twi and Ga for Ghana.

SME was underpinned by a philosophy of personal empowerment and guided by key behavioural theories including Social Cognitive Theory and Self-Regulation Theory to promote self-efficacy, individual care, equipping individuals with CMD with knowledge/skills to self-manage their conditions [62,63]. Using a person-centred self-management approach, it was important to draw attention to setting individual goals, information and education on CMD, feedback, engagement of family and therapeutic alliance between the facilitator and the patients. Table 3 provides a description of the SME design objectives, SME features and relevant SME sessions.

The target audience of the SME program were patients however health care providers were trained using the CREATE curriculum on how to deliver the SME programme to the patients.

Based on the Theoretical Domains Framework and Behaviour Change Technique Taxonomy and with the information steered by the guided principles, we identified key BCTs (active ingredients) for the SME programme (Table 4).

2. **Decision support:** Decision support tools included a comprehensive training package for HCPs, featuring CMD clinical algorithms, patient risk stratification tools, and guidelines for shared decision-making in resource-limited settings. The training package aimed to upskill frontline healthcare professionals with practical skills in CMD management, including decision-making tools, patient counselling techniques, and the effective use of clinical registries. This was co-developed with input from the needs assessment, scoping analysis findings and community engagement and involvement panel discussions. The training program for HCPs aimed to provide decision support through an improved knowledge base and clinical decision-making skills, expand knowledge of CMD and common MLTCs such as depression; the role of societal and cultural beliefs in disease management; appropriate use of technology and supporting resources; use of resources for care planning; and active listening. A total of 53 (29 in Mozambique, 9 in Ghana, and 15 in Kenya) health care workers were trained, remotely by a team of experts in Leicester, UK and each country's lead teams.

3. **Community linkages**: As part of this revised model of care, 55 community champions across the 3 countries [20 in Kenya, 10 in Ghana, and 25 in Mozambique], were trained by the team at the Leicester Diabetes Centre and in country teams, to raise awareness about cardiometabolic diseases and type 2 diabetes within their communities. This initiative aimed to enhance community engagement and support, promoting an environment where individuals are more informed and proactive about managing their health. By leveraging the impact and reach of community champions, we aimed to create a supportive network that reinforces self-management principles and promotes healthier lifestyles. Community champions trained can also be involved in health talks with the community in Churches, welfare groups, and around their businesses, among others, to raise awareness on diet, treatment, and medication.

4. **Organisation of health care:** It also involves encouraging collaboration, decision making and linkages with different health care providers and specialists for enhanced collaborative care.

5. **Clinical information system:** Given the lack of information systems in most sub-Saharan African settings, several discussions were held on how to set up a registry which is important in the clinical care of patients with MLTC. Currently, there are no decision support systems routinely in place across Ghana, Kenya and Mozambique. Most clinical assessments occurring at lower institutional levels are geared towards communicable diseases such as malaria, tuberculosis and HIV. However, discussions are on-going on how to have a common registry in place.

6. **Delivery system design. A streamlined referral pathway.** A functional referral system is essential in the continuum of care from the community to the primary health care providers and the secondary referral hospitals where patients can receive specialised care. At present, patients are not referred or seen by appropriate medical staff until complications

**Table 3. CREATE SME education design objectives, features and sessions content.**

| SME design objectives | SME features | SME sessions |
|---|---|---|
| To raise awareness on the importance of clinical medicine to address the cultural/individual belief system | Trusted and credible sources, evidence-based work developed by credible medical teams locally, or internationally (regarding T2DM and CVD).<br>Share examples of personal lifestyle changes (based on clinical medicine and behaviour change).<br>Local educator will present information on the current healthcare system and possible opportunities of free medication (educator note).<br>Local educator will be trained to acknowledge and address queries raised regarding traditional medicine. | T2DM and Heart Story<br>Medication myths activity (card game of individual quotes |
| To encourage healthy eating through a reduced salt, fat and carbohydrate diet? | Emphasis on affordable food providing examples of food relevant to the target audience.<br>Focus on creating lifestyle habits that rely less on carb-heavy, high salt and fat-based meals and lifestyle changes to traditional meals.<br>Focus on creating lifestyle habits that rely more on high fibre-based meals.<br>Build on self-efficacy and skills (link with Social Cognitive Theory). | Food activity game: Activity for educator to source local food and food packaging to bring to the session (fat, sugar, salt).<br>Food table (example of culturally relevant foods)<br>How to calculate the level of grams in the food portion (using spoons, instead of sugar cubes).<br>Sort the fat.<br>100 calorie game (walk it off). Fibre activity. |
| To help people understand the importance of high intensity exercise? | Acknowledge low intensity exercise but emphasise on high intensity physical activity. Provide examples on images.<br><br>Focus on creating lifestyle habits that are tailored to people's routine. | Physical activity continuum.<br><br>Recommendations of moderate vs vigorous physical activity.<br><br>Provide examples of physical activity (e.g., walking with friend, rope skipping, playing football). |
| To educate about CMD and CMD management | Focus less on text, and more on visuals to present information about CMD.<br>Provide very brief health messages using simple language. | Participant handbook mainly images.<br>Description of CMD displayed with images.<br>For the participant story, use images or drawing. |
| To promote user autonomy | Provide an environment for people to choose their own behaviour change, goals, action plan.<br>Provide an environment for people to reflect on questions (not expect the facilitator to be the expert).<br>Promote involvement and engagement in interactive activities and group discussions.<br>Promote the ethos of the programme that people are responsible for their own behaviour. | Self-management plan and provide time to reflect and complete.<br>Highlight the ethos and approach in the educator's training, and guided throughout the curriculum.<br>Set scene at the beginning of the session (e.g., interactive learning) |
| To promote intrinsic motivation and stress management | Use positive language throughout the programme and use key facilitation skills (listening, acknowledging people's stories, enable group discussion). | Relaxation square activity |
| To promote engagement | Consider the time and location of the training programme.<br>Consider the cost to attend the programme.<br>Promote SMEs on radio and other media. | |

from CMD arise. For the purpose of this study, the current systems for referral and care were mapped to understand existing systems within Ghana, Kenya and Mozambique. The first point of care for most patients in LMICs is usually a primary health care facility where there is no specialist.

## Discussion

Given the dearth of literature, we have developed a CoCCM which is applicable for people with multiple long-term conditions in LMICs. This is the first model of care for persons with multiple long-term conditions to be developed in LMICs. The strength of this model consists of its inclusivity, comprehensives and person-centeredness. The involvement of the

**Table 4. Key behavioural change techniques identified for the CREATE SME component.**

| Theoretic Domain Framework | Behavioural Change Techniques | Behavioural Change Technique Description |
|---|---|---|
| Knowledge | Health consequences | Offer information about the health benefits and consequences of CMD management. |
| | Feedback on behaviour | Provide an opportunity for people to receive feedback on the performance of their behaviour. |
| | Credible source | Provide information from credible and trustworthy sources. |
| Skills | Habit formation | Prompt people to take medication at a certain time. |
| | | Prompt people to come up with ideas on what food they can buy or eat. |
| Beliefs about capabilities/ optimism | Verbal persuasion to boost self-efficacy | Reassure in sessions that people can successfully make changes to their behaviour. |
| | Focus on past success | Make reference to people's past success in making changes. |
| Social influences | Social support and encouragement | Offer safe space for group discussions and promote encouraging language in sessions. |
| | Social support (practical) | Involve a partner or family in behaviour change at home. |
| Goals | Goal setting (behaviour) | Allow people to set individual goals (in relation to CMD behaviours). |
| | Review of behaviour goals | Allow people time to review their goal and make changes accordingly. |
| | Action planning | Prompt people to plan a detailed action for a specific behaviour, considering context, frequency, and duration among others. |
| | Problem solving | Provide an opportunity for people to work on an action plan. Enable people to develop the skills required to understand how to achieve their behaviour, by identifying potential barriers and ways to overcome them. |
| Environmental context and resources | Restructuring the physical environment | Prompt discussions about ways to change the physical environment. For example, reduce fast-foods. |
| Emotion | Reduce negative emotions | Address negative emotions that are associated with CMD. Allow time for people to share their concerns and focus on the positives of CMD management. |
| | | Provide relaxation skills. |
| Behavioural regulation | Self-monitoring of behaviour | Allow people to record and self-monitor the performed behaviour. |
| | | For example, ask people to record when they took medication. |

community members and persons living with CMD during development ensured that intervention components are culturally acceptable and are adapted to the local contexts of participating countries; hence enhancing model social acceptability and ownership during implementation. The model encourages health care providers to put the needs of the patient at the centre of care, while ensuring that patients are given self-management education on their MLTCs. The inclusion of community linkages strengthens the social support system, increasing individual patients and the community awareness of the management of MLTCs. Preliminary findings suggest that integrating self-management education with community linkages could significantly improve medication adherence and lifestyle modification among patients with CMD, aligning with evidence from prior CCM implementations that demonstrated an improvement in patient outcomes in LMICs [69].

## The collaborative chronic care model for the CREATE project

The CREATE CoCCM intervention model proposes integrating CCM with CoCM hence promises to be a powerful strategy in this context. The CCM brings on board evidence-based approaches to integrate care services for chronic illnesses, while CoCM emphasizes well-defined system components for high-quality care and fosters collaboration among healthcare providers. This approach concurs with the findings of a systematic review which showed an improvement in several aspects such as cardiometabolic risk reduction, diabetes self-management behaviours, and psychosocial well-being alongside individualised face-to-face intervention delivery in LMICs, with a special emphasis on low-income earners [70]. This holistic approach to intervention design echoes the results of a systematic review which showed that in comparison

to usual care, integrated care featuring at least two elements of Wagner's chronic care model resulted in a moderate improvement in systolic blood pressure among patients living with cardiometabolic multimorbidity [71]. By combining these frameworks, we aim to address the lack of primary healthcare integration, a major limitation to existing programs in LMICs. This integrated approach has the potential to reconcile ongoing debates on comprehensiveness, selectivity, fragmentation, and cohesion in sustainable and ethical integration of primary healthcare interventions [72]. In the development of the CREATE CoCCM, we identified six components that were required for the CREATE CoCCM which can be implemented in LMICs which included the following: 1) Self-Management support, 2) Clinical information system, 3) Delivery system design (streamlining the referral pathway), 4) Organisation of health care, 5) Decision support (which included health care provider training), and 6) Community linkages.

The CCM has been identified as the dominant organizational framework for chronic disease management in LMICs [73]. However, there are limitations related to community actors involvement, and self-management education support programs rarely address CMD; despite recommendations for a more interdisciplinary approach. These findings suggest a need to adapt the CCM for LMIC contexts, ensuring stronger community engagement and addressing the growing challenge of multimorbidity among patients. A meta-analysis has shown that a collaborative model that has a structured psychotherapy and family involvement elements are more effective in improving outcomes among patients with depression [74]. A systematic review in Ghana identified a significant challenge for care models like the CCM that rely on improved patients' self-management. The study revealed poor adherence to self-care behaviours among patients with type 2 diabetes. Using the WHO framework for adherence, the review highlighted various factors influencing self-care practices, including patient characteristics, sociodemographic and economic factors, disease-related factors, and limitations within the healthcare system. This underscores the need to address these multifaceted barriers when designing and evaluating interventions to improve self-management support for chronic disease management in LMICs.

The CoCCM is particularly well-suited for managing chronic medical diseases (CMD) due to its comprehensive, patient-centered approach that integrates various aspects of healthcare delivery. CMDs are the commonest clusters of multiple long-term conditions. This model addresses the unique challenges of CMD by emphasizing coordinated care, patient self-management, and the use of evidence-based guidelines, which are crucial for managing the complexities associated with CMD. The CoCCM's design facilitates improved patient outcomes and resource utilization, making it an effective framework for CMD management. The CoCCM has been applied to a variety of chronic conditions, including mental health disorders, chronic respiratory diseases, chronic kidney disease, and conditions affecting the elderly [47]. In the management of CMD, the CoCCM offers an effective framework for managing CMD through its integrated, team-based, and patient-centered approach. It addresses the complex and interrelated nature of conditions such as cardiovascular disease, type 2 diabetes, and obesity by promoting coordination among healthcare providers and emphasizing evidence-based, patient-focused care [75]. Key components of the CoCCM include team-based care, where multidisciplinary professionals collaboratively manage both metabolic and cardiovascular aspects of disease at the different levels of care [76,77]. The patient-centered approach enhances self-management and engagement, enabling patients to take an active role in decision making for their care [78]. The use of evidence-based guidelines ensures optimal treatment, to reduce cardiovascular risks [76]. Integrated care delivery improves continuity and coordination across healthcare settings, hence mitigating fragmentation of care [77,79]. The model's focus on prevention and early intervention supports timely identification and management of risk factors, reducing the onset of complications.

## Study strengths and limitations

The strengths of the development process of the intervention include engaging a multidisciplinary team across four countries, setting up community engagement panels and who gave their views and opinions on the model structure and context which further strengthened adaptation. On the other hand, the development process faced some limitations; the model development began during the COVID-19 pandemic, during which movement and meeting restrictions were in

place. As a result, the multi-country teams would have benefited from face-to-face meetings and consultations. Although all sites worked well together, distance may have created some challenges when developing the components.

We acknowledge that some components such as decision support and clinical information systems) were not consistently implemented across study sites and therefore were not the focus of this paper. Future work should more explicitly examine how contextual and structural factors influence the sustainability and scalability of the model.

## Conclusion

The CREATE CoCCM leverages evidence-based CMD care strategies to address gaps in healthcare delivery, emphasising cultural adaptation, self-management, and community engagement, with the potential to improve health outcomes in LMICs. Future research will focus on scaling the CREATE CoCCM through cluster RCTs to evaluate its impact on patient outcomes, healthcare utilization, and cost-effectiveness. Further adaptation studies will be conducted to explore integration into national health systems and sustainability in diverse LMIC contexts.

## Acknowledgments

The author team would like to thank the Community Engagement Panel's time and input to help culturally adapt the CREATE intervention. We also thank the local Educators, Community Champions and HCPs for being open to receiving the CREATE training. We would like to acknowledge the involvement of the IMPACT and EDEN team at LDC; the IMPACT team for helping to develop the SME programme and for developing and delivering the SME training to educators; the EDEN team for developing the training for Community Champions and HCPs.

## Author contributions

**Conceptualization:** Pamela Miloya Godia, Michelle Hadjiconstantinou, Samuel Seidu, Joyce Muhenge Olenja, Mary Coleman, Alfred Yawson, Filipe Dulce, Ana Mocumbi, Deborah Ikhile, Shabana Cassambai, Albertino Damasceno, Roberta Lamptey, Kamlesh Khunti.

**Data curation:** Pamela Miloya Godia, Rosa Weyula, Ana Mocumbi, Deborah Ikhile, Albertino Damasceno, Roberta Lamptey, Kamlesh Khunti.

**Formal analysis:** Pamela Miloya Godia, Michelle Hadjiconstantinou, Samuel Seidu, Joyce Muhenge Olenja, Albertino Damasceno, Roberta Lamptey, Kamlesh Khunti.

**Funding acquisition:** Pamela Miloya Godia, Michelle Hadjiconstantinou, Samuel Seidu, Deborah Ikhile, Shabana Cassambai, Roberta Lamptey, Kamlesh Khunti.

**Investigation:** Pamela Miloya Godia, Joyce Muhenge Olenja, Mary Coleman, Alfred Yawson, Kamlesh Khunti.

**Methodology:** Pamela Miloya Godia, Michelle Hadjiconstantinou, Rosa Weyula, Usagi Ememwa, Samuel Seidu, Peter Njoroge, Joyce Muhenge Olenja, George Nyadimo Agot, Jamin Avugwi, Mary Coleman, Alfred Yawson, Filipe Dulce, Joselia Chemane, Ana Mocumbi, Deborah Ikhile, Shabana Cassambai, Albertino Damasceno, Roberta Lamptey, Kamlesh Khunti.

**Project administration:** Pamela Miloya Godia, Rosa Weyula, Mary Coleman, Joselia Chemane, Celia Novela, Shabana Cassambai, Albertino Damasceno, Roberta Lamptey, Kamlesh Khunti.

**Resources:** Michelle Hadjiconstantinou, Deborah Ikhile, Albertino Damasceno, Kamlesh Khunti.

**Supervision:** Pamela Miloya Godia, Michelle Hadjiconstantinou, Rosa Weyula, Mary Coleman, Celia Novela, Roberta Lamptey, Kamlesh Khunti.

**Validation:** Rosa Weyula, Samuel Seidu, Celia Novela, Albertino Damasceno, Kamlesh Khunti.

**Visualization:** Pamela Miloya Godia, Michelle Hadjiconstantinou, Shabana Cassambai, Albertino Damasceno, Kamlesh Khunti.

**Writing – original draft:** Pamela Miloya Godia, Michelle Hadjiconstantinou, Rosa Weyula, Usagi Ememwa.

**Writing – review & editing:** Pamela Miloya Godia, Michelle Hadjiconstantinou, Rosa Weyula, Usagi Ememwa, Samuel Seidu, Peter Njoroge, Joyce Muhenge Olenja, George Nyadimo Agot, Jamin Avugwi, Mary Coleman, Alfred Yawson, Celia Novela, Deborah Ikhile, Shabana Cassambai, Albertino Damasceno, Roberta Lamptey, Kamlesh Khunti.

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
