## [Decision Letter · Decision Letter 0]

3 Sep 2025

Dear Dr. Pamela Miloya Godia,

We look forward to receiving your revised manuscript.

Kind regards,

Paolo Magni

Academic Editor

PLOS ONE

Journal Requirements:

“I have read the journal's policy and the authors of this manuscript have the following competing interests:

KK has acted as a consultant, speaker or received grants for investigator-initiated studies for Astra Zeneca, Boehringer Ingelheim, Bayer, Novo Nordisk, Sanofi-Aventis, Servier, Lilly and Merck Sharp & Dohme, Oramed Pharmaceuticals, Pfizer, Roche, Daiichi-Sankyo, Applied Therapeutics, Embecta, Amgen, Bristol Myers Squibb and Nestle Health Science. KK is supported by the National Institute for Health Research (NIHR) Applied Research Collaboration East Midlands (ARC EM), NIHR Global Research Centre for Multiple Long-Term Conditions, NIHR Cross NIHR Collaboration for Multiple Long-Term Conditions,NIHR Leicester Biomedical Research Centre (BRC) and the British Heart Foundation (BHF) Centre of Excellence.

SS is in receipt of speaker honoraria from AstraZeneca, Boehringer Ingelheim, Janssen, Lilly, MSD, Abbott, Novo Nordisk, SB Communications, OmniaMed Communications, Roche, Napp Pharmaceuticals, NB Medical and Amgen; advisory board honoraria from AstraZeneca, Lilly, Boehringer Ingelheim, Janssen, Abbott, MSD, Novo Nordisk, Takeda and Sanofi; educational grants from Boehringer Ingelheim, Lilly, Novo Nordisk, and Takeda; and conference registration and subsistence from Boehringer Ingelheim, Janssen, Lilly, Novo

Nordisk, Abbott and Takeda.

RL has received speaker honoraria, advisory board or educational grants or research funding: Novo Nordisk, Boehringer Ingelheim, AstraZeneca, Sanofi.

SC are supported by the National Institute for Health Research (NIHR) Global Health Research Group for Cardiometabolic Disease Research in Africa Partnership (CREATE).

SC, DI, are supported by the NIHR Applied Research Collaboration East Midlands (ARC EM), NIHR Global Research Centre for Multiple Long-Term Conditions.

PG, MH, RW, UE, PJ, JO, GA, JA, MC, AY, DF, AM, JC, CN, AD have declared no competing interests exists.”

5. We note that Figures 1 and 2 in your submission contain copyrighted images. All PLOS content is published under the Creative Commons Attribution License (CC BY 4.0), which means that the manuscript, images, and Supporting Information files will be freely available online, and any third party is permitted to access, download, copy, distribute, and use these materials in any way, even commercially, with proper attribution. For more information, see our copyright guidelines: http://journals.plos.org/plosone/s/licenses-and-copyright.

a. You may seek permission from the original copyright holder of Figures 1 and 2 to publish the content specifically under the CC BY 4.0 license.

Additional Editor Comments:

The paper offers some interesting information and proposes a relevant model for chronic care of cardiometabolic diseases in low/middle income countries.

However, several detailed comments raised by the Reviewers need to be addressed in full.

Reviewers' comments:

Reviewer's Responses to Questions

**Comments to the Author**

1. Is the manuscript technically sound, and do the data support the conclusions?

Reviewer #1: Partly

Reviewer #2: Yes

2. Has the statistical analysis been performed appropriately and rigorously?

Reviewer #1: N/A

Reviewer #2: N/A

3. Have the authors made all data underlying the findings in their manuscript fully available?

Reviewer #1: No

Reviewer #2: No

4. Is the manuscript presented in an intelligible fashion and written in standard English?

Reviewer #1: Yes

Reviewer #2: No

Reviewer #1: The manuscript provides a timely and relevant overview of cardiometabolic diseases (CMDs), particularly highlighting the disproportionate burden borne by low- and middle-income countries (LMICs). The authors propose an adaptation of the Chronic Care Model (CCM) as a potential response, with an emphasis on person-centred and community-orientated care.

A key strength of the study lies in its effort to triangulate multiple sources of data (from a literature review to community engagement), which are then synthesised into a proposed model. The emphasis on preserving core CCM principles while adapting the model to LMIC contexts is commendable.

To further strengthen the manuscript, it may be helpful to situate this work more clearly within the broader body of literature that has sought to revise or adapt the traditional CCM. Several recent iterations have moved toward greater collaboration and integration across sectors, and referencing these could help clarify how the proposed Collaborative Chronic Care Model (CoCCM) builds upon, or diverges from, existing efforts.

While the focus on CMDs is well-argued, it would be valuable to more explicitly demonstrate how the CoCCM offers unique utility for CMD management compared to its application for other chronic conditions. Clarifying this would enhance the specificity and relevance of the proposed model.

There appears to be some misalignment between the stated knowledge gap, namely, the lack of evidence around the implementation and scale-up of collaborative care models in LMICs, and the study’s stated aim to develop a person-centred model of care for CMD management. Clarifying how the study responds to this gap may help improve the coherence of the paper.

The framing of LMICs (and Africa in particular) could be strengthened. At times, the manuscript presents a high-income versus low-income dichotomy, which may unintentionally reinforce deficit-based narratives. It also overlooks evidence showing that cardiovascular disease (CVD) mortality has plateaued, or even increased, in some high-income countries.

A more balanced approach would be to highlight both the ongoing challenges, the local innovations and gains in CMD management seen in specific African countries such as Kenya, Ghana, and Mozambique (which the paper later references). This would offer a more nuanced and empowering perspective, rather than generalising across the entire African continent or all LMICs.

It may also be helpful to reconsider the use of the term “sub-Saharan Africa”, as it has been critiqued in scholarly literature for its lack of specificity and its problematic historical associations.

The manuscript could benefit from more explicit articulation of how the proposed CoCCM was shaped by the specific needs, characteristics, and resources of people living with CMD in the LMIC (or African) context. While the literature review and data collection are comprehensive, it is not entirely clear what was learned about the unique needs, characteristics and resources across the study settings and how these inputs were translated into design elements of the model. More detail on this process would enhance the model’s contextual relevance and practical applicability.

Finally, by focusing mainly on individual behaviour change as the key aspect of the CoCM model, the authors risk advocating for a care model that ignores how structural factors constrain or facilitate individual agency.

Reviewer #2: This manuscript presents the development of a health care model for people with multiple chronic cardiometabolic diseases (CMD/MLTC) in sub-Saharan Africa. The authors clearly articulate the study’s aim, and ethical clearance was appropriately obtained from the relevant committees in Kenya, Ghana, and Mozambique.

The development of the model was informed by a systematic review, qualitative synthesis, and a needs assessment with stakeholders. The resulting CREATE intervention comprises five key components: a self-management education (SME) programme, a training package for educators to deliver the SME, a training package to upscale health care providers and community champions in CMD, a decision making tool for a data registry, and an improved referral pathway.

Although the authors have stated that all information is included in the manuscript and the supporting information files, I could not find any access links to the supporting files. I believe that resources such as guides for the SME sessions, community linkage training materials, and shared decision-making guidelines, are of interest in the context of this manuscript. Inclusion of those resources in the supporting information files should be considered.

The manuscript is generally well-structured and linguistically sound. The following minor revisions are recommended to accept the manuscript for publication. All cited pages below refer to the manuscript page at the bottom of each page and not the pdf page.

1. To my understanding, the chronic conditions focused on by this program are T2DM, hypertension, cholesterol, and cardiovascular disease. I suggest describing in the manuscript why the model was limited to these four chronic conditions.

2. Table 3: The alignment between SME features and SME sessions in Table 3 is unclear. For example, SME feature "Focus on creating lifestyle habits that are tailored to people's routine" is in the same row as SME sessions "Recommendations of moderate vs vigorous", which seem unrelated to each other. I suggest modifying the data presentation to avoid misunderstanding. Some options: Option 1. nested table (instead of aligning features and sessions row by row, nest the features and sessions under the same objective: one row per objective, list all features in the objective in the same row in column 2, list all sessions under the objective in the same row in column 3). Option 2. bulleted concept map (diagram). Other forms may be utilized according to how the authors see fit.

3. Kindly specify who the audience for the SME program was: HCP, patients, or both. (page 16-17)

4. The six components of CREATE CoCCM in the article were 1) Self-Management support, 2) Clinical information system, 3) Delivery system design (streamlining the referral pathway), 4) Organisation of health care, 5) Decision support (which included health care provider training), and 6) Community linkages; these do not match the components in the results section of the Abstract. Consider streamlining for clarity.

5. Please check the manuscript and amend some grammatical and typographical errors, including, but not limited to:

- consistency with American or British spelling (British spelling was used for the phrase "Randomised Control Trials", whereas other parts of the manuscript uses American English).

- inaccurate use of capital letters where they are not needed e.g. "Randomised Control Trials" on page 8 and "Systolic Blood Pressure" on page 25.

- consistency with abbreviations; some terms' abbreviations have been specified but unabbreviated terms are used again in the later parts of the manuscript. For example, "Multiple long-term conditions (MLTC)" has been cited on page 8, but is inconsistently written as the unabbreviated form (pages 8, 9, 10) and abbreviated form (page 24). Similarly for "Chronic Care Model (CCM)" and "Collaborative care models".

- grammatical errors such as "traditional high calorific calories" -> "high calorific value" or "high calorie foods".

**Do you want your identity to be public for this peer review?** For information about this choice, including consent withdrawal, please see our Privacy Policy

Reviewer #1: **Yes:** Kenneth Yakubu

Reviewer #2: No

---

## [Author Response · Author response to Decision Letter 1]

5 Jan 2026

Response to Reviewer Comments

Reviewer #1: The manuscript provides a timely and relevant overview of cardiometabolic diseases (CMDs), particularly highlighting the disproportionate burden borne by low- and middle-income countries (LMICs). The authors propose an adaptation of the Chronic Care Model (CCM) as a potential response, with an emphasis on person-centred and community-orientated care. A key strength of the study lies in its effort to triangulate multiple sources of data (from a literature review to community engagement), which are then synthesised into a proposed model. The emphasis on preserving core CCM principles while adapting the model to LMIC contexts is commendable.

Reviewer comment #1: To further strengthen the manuscript, it may be helpful to situate this work more clearly within the broader body of literature that has sought to revise or adapt the traditional CCM. Several recent iterations have moved toward greater collaboration and integration across sectors and referencing these could help clarify how the proposed Collaborative Chronic Care Model (CoCCM) builds upon, or diverges from, existing efforts.

Author response:

We thank the reviewer for this valuable suggestion. The literature review section has been updated to include a detailed discussion of adaptations made to the traditional Chronic Care Model (CCM), highlighting how our proposed CoCCM builds on existing revisions. (Pages 8–9 of the manuscript).

Adaptations to the traditional chronic care model

Recent literature highlights substantial adaptations to the traditional CCM, aimed at strengthening cross-sector collaboration and integration in chronic disease management. These revisions emphasize multidisciplinary teamwork, leveraging technology, systemic reforms, and patient-centered and holistic care, ensuring a more holistic and coordinated approach.

Multidisciplinary teams (MDTs) integration now forms the backbone of the revised CCM, incorporating pharmacists, nurses, general practitioners, and psychologists to deliver comprehensive, patient-centered care (1,2). The inclusion of pharmacists improves medication adherence and reduces errors, enhancing outcomes while reducing healthcare costs (1). The integration of digital health tools, eHealth platforms, and interoperable information systems strengthens communication and data sharing across sectors (2,3). Digital solutions enhance coordination of patient care by enabling proactive monitoring, population-level data analysis, and have also been critical in the integration of mental and physical health management (4). The eHealth Enhanced CCM (eCCM) highlights the value of virtual communities and feedback loops in boosting patient engagement (5).

Recent literature calls for policy reforms, shared care plans, and interprofessional training to standardize collaboration and clarify roles and responsibilities within MDTs (2). Revisions of the CCM include integration of physical, emotional, and social dimensions of health, hence fostering individualized care plans that improve satisfaction and outcomes. The Expanded Chronic Care Patient–Professional Partnership Model (E2C3PM) is a novel integrated model that leverages patient experience and knowledge, mediation, therapeutic education, and partnership to empower patients, rebalance power dynamics, and foster a more comprehensive and collaborative approach to chronic disease management, leading to improved care integration and outcomes (6). Community integration across healthcare and social sectors improves care coordination for complex patient needs, reducing exclusion and supporting continuity of care. Applications of the CCM in contexts such as the Veteran’s Health and correctional systems demonstrate its adaptability to diverse populations and institutional settings (7). Community-based organizations and primary care nurses are increasingly recognized as key actors in supporting self-management and extending care beyond clinical settings (8).

Despite these advancements, barriers such as resistance to change, lack of standardized protocols, and limited adoption of eHealth and patient-centred and partnership approaches persist (3). Effective implementation requires sustained innovation, organizational cultural shifts, and adaptation to local contexts to fully optimize chronic care delivery.

Reviewer comment #2. While the focus on CMDs is well-argued, it would be valuable to more explicitly demonstrate how the CoCCM offers unique utility for CMD management compared to its application for other chronic conditions. Clarifying this would enhance the specificity and relevance of the proposed model.

Apart from its use in management cardiometabolic diseases, how can we explicitly demonstrate the application of the Collaborative Chronic Care Model (CoCCM) for other chronic conditions.

Author response:

We agree with this comment and have elaborated in the discussion section (page30) how the CoCCM is specifically suited for cardiometabolic disease (CMD) management, in comparison to other chronic medical conditions.

The CoCCM is particularly well-suited for managing chronic diseases due to its comprehensive, patient-centered approach that integrates various aspects of healthcare delivery. CMDs are the commonest clusters of multiple long-term conditions. This model addresses the unique challenges of CMD by emphasizing coordinated care, patient self-management, and the use of evidence-based guidelines, which are crucial for managing the complexities associated with CMD. The CoCCM's design facilitates improved patient outcomes and resource utilization, making it an effective framework for CMD management.

The CoCCM has been applied to a variety of chronic conditions, including mental health disorders, chronic respiratory diseases, chronic kidney disease, and conditions affecting the elderly(9). In the management of cardiometabolic disease, the CoCCM offers an effective framework for managing cardiometabolic diseases through its integrated, team-based, and patient-centered approach. It addresses the complex and interrelated nature of conditions such as cardiovascular disease, type 2 diabetes, and obesity by promoting coordination among healthcare providers and emphasizing evidence-based, patient-focused care(10).

Key components of the CoCCM include team-based care, where multidisciplinary professionals collaboratively manage both metabolic and cardiovascular aspects of disease at the different levels of care (11,12). The patient-centered approach enhances self-management and engagement, enabling patients to take an active role in decision making for their care(13). The use of evidence-based guidelines ensures optimal treatment, to reduce cardiovascular risks (11). Integrated care delivery improves continuity and coordination across healthcare settings, hence mitigating fragmentation of care (12,14). The model’s focus on prevention and early intervention supports timely identification and management of risk factors, reducing the onset of complications.

Reviewer comment #3. There appears to be some misalignment between the stated knowledge gap, namely, the lack of evidence around the implementation and scale-up of collaborative care models in LMICs, and the study’s stated aim to develop a person-centred model of care for CMD management. Clarifying how the study responds to this gap may help improve the coherence of the paper.

Author response:

We clarified the knowledge gap on Page 10, emphasizing that our study responds to the lack of evidence on adaptation, scalability, and sustainability of collaborative care models in LMICs

There exists knowledge gap in the adaptation, scalability and sustainability of chronic care models that have been successfully implemented to fit the LMIC context in relation to constraints related to the health system, infrastructure and social and cultural factors. To date, evidence on the implementation of collaborative care models emanate from high income countries with little adaptation to LMIC settings (15–17). There is therefore a lack of evidence on how to effectively adapt and scale interventions in diverse LMIC settings, which is crucial for addressing common chronic conditions such as cardiometabolic diseases and mental health (9,16). Many LMICs face health system constraints such as inadequate infrastructure, shortage of trained health professionals, limited access to essential medications and diagnostic tools (18), which may pose significant challenges in the implementation of collaborative care models. This calls for the need to develop and test novel low-cost approaches which are tailored to resource-limited settings to address these disparities (18). There is also knowledge gap in the understanding the use of digital technology, health care provider training, and supervisory structures including how to effectively integrate mid-level practitioners and community health workers necessary for effective chronic care management (19).

Reviewer comment #4. The framing of LMICs (and Africa in particular) could be strengthened. At times, the manuscript presents a high-income versus low-income dichotomy, which may unintentionally reinforce deficit-based narratives. It also overlooks evidence showing that cardiovascular disease (CVD) mortality has plateaued, or even increased, in some high-income countries.

Author response:

Thank you for this comment. We have provided a reference stating the increase and plateau of CVD in high income countries.

This has been noted on Page 4 of the manuscript.

However, data published in the World Health Organization Mortality Database indicate that the rate of decline in mortality due to cardiovascular disease in some high-income countries has slowed down or plateaued particularly for persons aged 35-74yrs; for example, North America (USA and Canada), mortality rates due to CVD have increased in recent years (20). In Switzerland, while CVD mortality continues to decrease in most of the population, specific age groups, such as those aged 60-74, have seen a plateau in stroke and coronary heart disease mortality rates. The plateau could be attributed to several factors such as lack of regular physical activity, unhealthy diets, high alcohol consumption, persistent smoking and social inequalities among different segments of the population (21).

Reviewer comment #5. A more balanced approach would be to highlight both the ongoing challenges, the local innovations and gains in CMD management seen in specific African countries such as Kenya, Ghana, and Mozambique (which the paper later references). This would offer a more nuanced and empowering perspective, rather than generalising across the entire African continent or all LMICs.

Author response:

On-going challenges and local innovation has been clarified on pages 10-11 of the manuscript.

Challenges and innovations in CMD management

Management of CMD in Kenya, Ghana and Mozambique face challenges which include health system factors, social and cultural factors such as lack of access to early diagnostic tools, lack of access to essential medicines and frequent stockouts in public hospitals, use of traditional medicine and disproportionate distribution of health care providers between urban and rural settings. For instance, Kenya has reported an increase in the prevalence of hypertension, but only 4% of the patients under treatment achieve control(22). With regards to early detection of diabetes, more that 88% of Kenyans have never had their blood glucose tested, leading to late diagnosis (22). Less than 50% of hospitals in Kenya have the equipment and trained staff required for essential diagnostic procedures like electrocardiography and ultrasonography and only 49% of hospitals have a complete stock of necessary medications, with frequent stock-outs reported (18). Similarly in Ghana challenges include inadequate infrastructure, workforce shortages, fragmented coordination of care, high costs of care and out-of-pocket expenditure, and governance inefficiencies for chronic disease management, reliance on traditional medicine, cultural beliefs and delayed care seeking (23,24)

Promising innovations in the management of CMDs in Kenya, Ghana and Mozambique include community-based lifestyle interventions which have reported significant reductions in metabolic syndrome markers such as improved diet and exercise adherence(25), and task-sharing models focusing on community health worker-led screening and linkage programs leveraging on digital technology (26). The Akoma Pa digital, faith-based pilot intervention tested in Ghana has reported higher patient follow-up rates, improved blood pressure and glycemic control, and demonstrated favorable cost-effectiveness(27). Mozambique has adapted community-based medication delivery models from successful HIV care programs to improve chronic disease management (28).

Reviewer comment #6. It may also be helpful to reconsider the use of the term “sub-Saharan Africa”, as it has been critiqued in scholarly literature for its lack of specificity and its problematic historical associations.

Author response:

This is correct.

The use of the term sub-Saharan Africa throughout the manuscript has been reviewed since this also excludes North Africa from the rest of the continent. Where reference is made to the three study countries (Kenya, Ghana and Mozambique) this has been specified to refer to the three countries, but where existing literature focus is on SSA, this has been maintained.

Reviewer comment #7. The manuscript could benefit from more explicit articulation of how the proposed CoCCM was shaped by the specific needs, characteristics, and resources of people living with CMD in the LMIC (or African) context. While the literature review and data collection are comprehensive, it is not entirely clear what was learned about the unique needs, characteristics and resources across the study settings and how these inputs were translated into design elements of the model. More detail on this process would enhance the model’s contextual relevance and practical applicability.

Author response:

Thank you for this comment.

We have expanded the manuscript to outline how the proposed CoCCM was shaped by the specific needs, characteristics, and resources of people living with CMD in the LMIC (or African) context.

This is included on pages 11-12 of the manuscript

The qualitative interviews and stakeholder workshops revealed several key unmet needs among people living with cardiometabolic diseases (CMDs). These included low awareness of disease signs and symptoms, resulting in poor health-seeking behaviour and late diagnosis. Patients also cited limited access to essential medicines, high costs of laboratory monitoring, and unhealthy dietary practices. Care-seeking was often initiated by an acute episode such as collapse. Many patients wanted to treat CMDs as acute illnesses and often asked if they could discontinue taking medication if they got symptom relief. Additionally, there was a notable lack of health education and suitable educational materials to support disease management and awareness. These perceptions informed the design of the SME and the community linkages components of the intervention. With regards to available resources, there were existing national policies and guidelines in the three countries which provided a framework for intervention design. However, there were notable gaps in the implementation and referral practices. Stakeholders also cited negative health care provider attitude, provision of fragmented care and the need to train healthcare providers to improve communication among themselves and with patients. At the community level, the following community structures were identified for intervention delivery: patient support groups, community health workers and volunteers, community leaders, women groups, church groups and keep-fit clubs.

With regards to intervention delivery stakeholders and persons with lived experiences noted that the intervention should be designed in such a way that it is delivered at certain venues, days, times to address challenges around transport and work-life commitments. It also transpired that literacy was a unique need across the study settings

---

## [Decision Letter · Decision Letter 1]

22 Feb 2026

Development of a collaborative chronic care model for management of cardiometabolic disease in low- and middle-income countries

PONE-D-25-22410R1

Dear Dr. Godia,

We’re pleased to inform you that your manuscript has been judged scientifically suitable for publication and will be formally accepted for publication once it meets all outstanding technical requirements.

Kind regards,

Paolo Magni

Academic Editor

PLOS One

Additional Editor Comments (optional):

The paper has been improved and all reviewers' comments have been addressed.

Reviewers' comments:

Reviewer's Responses to Questions

**Comments to the Author**

Reviewer #1: All comments have been addressed

Reviewer #2: All comments have been addressed

2. Is the manuscript technically sound, and do the data support the conclusions?

Reviewer #1: Yes

Reviewer #2: Yes

3. Has the statistical analysis been performed appropriately and rigorously?

Reviewer #1: N/A

Reviewer #2: N/A

4. Have the authors made all data underlying the findings in their manuscript fully available?

Reviewer #1: Yes

Reviewer #2: Yes

5. Is the manuscript presented in an intelligible fashion and written in standard English?

Reviewer #1: Yes

Reviewer #2: Yes

Reviewer #1: (No Response)

Reviewer #2: Thank you for the opportunity to review the revised version of the manuscript. I appreciate the authors' careful and thorough responses to the comments raised in the initial review.

The following revisions, along with the revisions made in response to Reviewer #1, have substantially strengthened the manuscript:

- The rationale for focusing on the selected cardiometabolic conditions has been clarified

- The format of Table 3 has been improved

- The clarification of the SME target audience enhances reader understanding of the programme

- The alignment between the intervention components in the abstract and the main text has been addressed

- The manuscript has been edited for consistency and grammatical accuracy

I have no further substantive comments and consider the manuscript suitable for publication in its current form.

**Do you want your identity to be public for this peer review?** For information about this choice, including consent withdrawal, please see our Privacy Policy

Reviewer #1: **Yes:** Kenneth Yakubu

Reviewer #2: No

---

## [Editor Report · Acceptance letter]

PONE-D-25-22410R1

PLOS One

Dear Dr. Godia,

I'm pleased to inform you that your manuscript has been deemed suitable for publication in PLOS One. Congratulations! Your manuscript is now being handed over to our production team.

Kind regards,

on behalf of

Prof. Paolo Magni

Academic Editor

PLOS One